# Capturing chemical intuition in synthesis of metal-organic frameworks

Seyed Mohamad Moosavi [1], Arunraj Chidambaram [1], Leopold Talirz[1,2], Maciej Haranczyk [3],
Kyriakos C. Stylianou [1] & Berend Smit [1]

We report a methodology using machine learning to capture chemical intuition from a set of (partially) failed attempts to synthesize a metal-organic framework. We define chemical intuition as the collection of unwritten guidelines used by synthetic chemists to find the right synthesis conditions. As (partially) failed experiments usually remain unreported, we have reconstructed a typical track of failed experiments in a successful search for finding the optimal synthesis conditions that yields HKUST-1 with the highest surface area reported to date. We illustrate the importance of quantifying this chemical intuition for the synthesis of novel materials.

[1] Laboratory of Molecular Simulation (LSMO), Institut des Sciences et Ingénierie Chimiques, Valais, École Polytechnique Fédérale de Lausanne (EPFL), Rue de l'Industrie 17, CH-1951 Sion, Switzerland. [2] Theory and Simulation of Materials (THEOS), Faculté des Sciences et Techniques de l'Ingénieur, École Polytechnique Fédérale de Lausanne (EPFL), Station 9, CH-1015 Lausanne, Switzerland. [3] IMDEA Materials Institute, C/Eric Kandel 2, 28906 Getafe, Madrid, Spain. Correspondence and requests for materials should be addressed to B.S. (email: berend.smit@epfl.ch)

Since two decades ago, when metal-organic frameworks (MOFs) emerged as a versatile class of materials for variety of applications, the chemistry and applications of MOFs have been the subject of a large body of research across several disciplines[1,2]. MOFs were described by the concept of reticular chemistry as materials composed of structural building blocks assembled on a net[3]. The scientific excitement about MOFs originates in the fact that by modifying the building blocks, i.e., changing the metal nodes or organic ligands, MOFs can be tuned for a given application. Therefore, in principle, the number of possible materials is infinitely large; however, since synthesis and optimisation of these materials can be time-consuming and laborious[4], only a fraction of them have ever been synthesised.

The synthesis of MOFs involves the self-assembly of the structural building blocks (known as secondary building blocks (SBUs)) in a 3D periodic network. However, our understanding of the self-assembly procedure, i.e., the kinetics and energetics of framework bond formation, nucleation, and crystal growth, has remained too limited to guide the synthesis of these materials. Specifically, since diverse and numerous chemistries exist in MOFs, even the known synthesis conditions for one MOF are typically not transferable to new MOFs, and accordingly, this has prevented chemists to draw a general synthetic route for these materials. The parameters for a typical MOF synthesis include the selection of solvents and their composition, temperature, and reaction time, etc. Considering each parameter as a variable, one needs to probe the high-dimensional chemical space constructed by these variables to find sets of synthesis conditions leading to the formation and crystallization of the desired MOF. Without any prior knowledge, one could envision a brute force approach and perform, say, a large grid search of the chemical space using robotic synthesizers. The cost of this approach increases exponentially with the number of variables, e.g., testing only ten choices for a space of nine variables requires a billion experiments. With such poor statistics, one may wonder how so many MOFs could have been synthesized? Clearly, the fact that thousands of MOFs have been synthesized[5] indicates that chemists have been able to beat brute force statistics by orders of magnitude. Given that at present there are at best some empirical guidelines, one can argue that their selection of experimental conditions must have been positively biased by the chemical intuition that synthetic groups have acquired. The aim of this work is to develop a systematic approach to capture this chemical intuition using machine learning. Recently, machine learning is starting to be applied to chemical synthesis[6–12]. Most of these efforts focus on predicting the outcome of a specific reaction. For instance, Raccuglia et al.[10] proposed and tested successfully the synthesis of a material by machine learning failed experiments using decades of old notebooks of chemical synthesis. Ahneman et al.[8] trained a random forest to predict the performance of the Pd-catalyzed Buchwald-Hartwig reaction.

For MOF synthesis, the ligands and metal nodes are in most cases sufficiently simple or even commercially available that their synthesis is often not the bottleneck. Most time and effort are spent in finding the optimal conditions for the ligands and metal nodes to self-assemble into crystals. While publications typically report only the most successful synthesis conditions, the chemical intuition is built from all experiments, in particular, the substantial number of partially successful and failed experiments. Hence, in this work, we start with reconstructing these unreported data for a prototypical MOF in the search for the optimal synthesis conditions. By analysing the generated data using machine learning, we capture and quantify the chemical intuition that researchers develop in their search for these optimal conditions. Later, we show the significance of this quantified intuition

in synthesis of another MOF where we show the intuition can be transferred while the detailed chemistry is not transferable.

## Results

**Synthesis and optimisation of the surface area of HKUST-1.** To illustrate our methodology, we focus on a real-life example of MOF synthesis. HKUST-1 (Hong Kong University of Science and Technology, also known as Cu-BTC, which is made up of Copper ions and 1,3,5-benzenetricarboxylic acid (BCT)) is a well-studied MOFs that has been synthesized by a large number of different groups (see the Supplementary Note 9 for a summary of the different synthesis methods)[13–16]. Although all groups report high-quality powder X-ray diffraction patterns, the different samples show Brunauer–Emmett–Teller (BET) surface area ranging from ~300 to ~2000 $m^2 g^{-1}$[13]. The comparison of the different synthesis conditions (see Supplementary Note 9 for details) shows that they differ mainly in solvent composition (e.g., mixtures of DMF, water, different alcohols, and others), temperature (25°–180 °C), and methods (e.g., conventional heating, microwave, electrochemistry, mechanochemistry, ultrasonic, etc.). At present, we lack the knowledge to explain why there are such differences in the BET surface areas, yet from a practical point of view it is important to obtain this material with the highest surface area [17].

One can safely state that this body of work on HKUST-1 involves hundreds if not thousands of experiments, of which only the successful conditions have been published. In this work, we aim to make the case that important and useful information can be obtained, if these groups would also have published their (partially) failed experiments. We use a robotic synthesis procedure to efficiently regenerate part of the failed and partially successful experiments that have been performed in the course to synthesize this material. Using a robotic synthesis platform improves the reproducibility of the generated data. Our robotic synthesizer uses microwave heating and the synthetic procedure involves selecting the setting of 9 different parameters that fully specify the synthesis conditions. Hence, a particular experimental condition can be described as a point in a 9-dimensional (chemical) space (Supplementary Table 1 and Supplementary Note 1). We have selected the ranges of synthesis conditions such that they include those solvents and temperatures that have been reported as successful in the literature, but not necessarily using microwave heating. Our robot can carry out 30 reactions per cycle, where a cycle is completed typically within one day. A simple grid search to explore all possible experimental conditions would require of the order of $10^9$ robot cycles, which illustrates the need of this chemical intuition, or in our case, in which we impose a lack of intuition, enhanced sampling techniques.

In the case of HKUST-1, several quite different successful synthesis conditions have been reported. Since the location of these sets of conditions are not known a priori, and for instance, might be clustered in relatively small islands in the high-dimensional space, pinpointing them is genuinely non-trivial. Simple gradient-based algorithms are discarded here due to the high probability of winding up in a local optimum. Genetic algorithms (GAs) have proven to be a robust global optimization algorithm for searching such a complex space[18,19]. The optimisation strategy in a GA is inspired by natural selection, nature's optimisation strategy. The 9-dimensional synthesis vector takes the role of the chromosome, carrying the synthesis variables as its genes, which are evolved via selection, crossover, and mutation (see Supplementary Note 1 for details). Only the mutated genes of successful parents are transferred to the next generation, thus optimizing the synthesis conditions generation by generation.

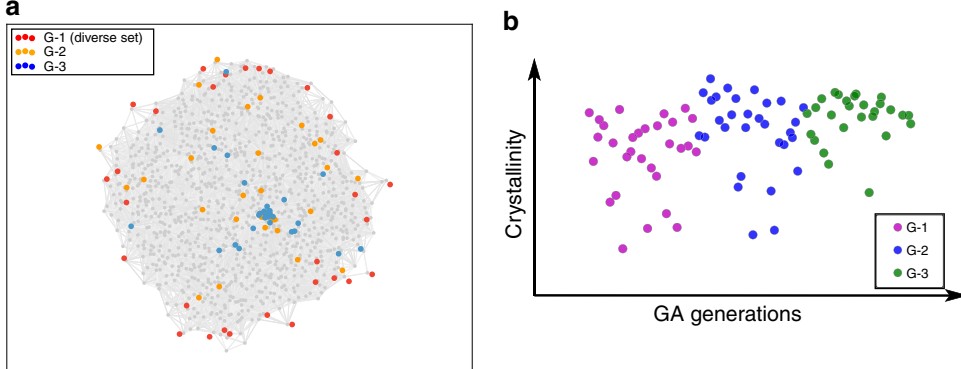

**Fig. 1** Optimisation of synthesis condition of Cu-HKUST-1. **a** Multidimensional scaling projection of the 9-D space of parameters onto a 2-D plane. In this representation, similar conditions are plotted close to each other, and connected if they have normalized pairwise distance below 0.1. Grey dots visualize the extent of the entire bounded (chemical) space, represented by mapping the set of 1000 most diverse synthesis conditions obtained from the MaxMin method. The red dots are the first 30 of this set which are used for the first experiments (G-1), the orange and blue dots mark the second (G-2) and third (G-3) generations obtained from the first via the genetic algorithm (GA). **b** Progress in crystallinity during GA optimization. The colour of dots indicates the generation in the GA. (HKUST = Hong Kong University of Science and Technology)

**Table 1 BET surfaces and the corresponding synthesis conditions of the five samples with the highest crystallinity**

| Sample | BET [$m^2 g^{-1}$] | $H_2O$ [ml] | DMF [ml] | EtOH [ml] | MeOH [ml] | iPrOH [ml] | Reactants ratio | Temperature [°C] | Microwave power [W] | Reaction time [mins] |
|---|---|---|---|---|---|---|---|---|---|---|
| 1 | 367 | 0.5 | 0.0 | 5.0 | 0.0 | 1.0 | 0.9 | 120 | 174 | 58 |
| 2 | 526 | 0.5 | 1.0 | 0.0 | 4.0 | 0.0 | 1.8 | 176 | 246 | 44 |
| 3 | 935 | 0.0 | 4.5 | 0.0 | 0.0 | 0.0 | 1.8 | 123 | 200 | 7 |
| 4 | 1596 | 0.0 | 4.0 | 0.0 | 0.0 | 2.0 | 0.8 | 200 | 240 | 60 |
| 5 | 2045 | 0.5 | 2.5 | 2.0 | 0.0 | 0.0 | 1.5 | 140 | 200 | 20 |

We start the search for the optimal synthesis conditions without any chemical intuition, i.e., all components of the 9-dimensional synthesis vector are considered equally important. The first run aims to cover the experimental space as widely as possible, using the MaxMin method[20], to obtain the set of 30 most diverse synthesis conditions. Figure 1a shows these conditions in a multidimensional scaling (MDS) projection. MDS-plots visualize the similarity between individuals in a dataset[21]. In this study, the Euclidian distance of normalized variables measures the similarity between synthesis trials. In Fig. 1a, similar synthesis trials are mapped close to each other while dissimilar experiments are far from each other on the map (see Method section for details). As expected, but not intuitively obvious, in such a high-dimensional space the most diverse set is located at the edges. The synthesis is attempted for each of the conditions, and the crystallinity and phase purity of the resulting samples are analyzed. Using those metrics for the objective function, we evolve the second generation and perform synthesis for all 30 new conditions. We measure crystallinity, phase purity, and BET surface area of these samples, and combine those metrics for the objective function for the third generation (see Supplementary Note 1 for details).

Figure 1b shows the progress in crystallinity over the three generations of experiments. The GA generations contain several different synthesis conditions that yielded samples with ideal powder X-ray diffraction pattern and phase purity. For highly crystalline samples in each generation, we determined the BET surface area (see Table 1), and, not surprisingly, find a wide range of BETs, including the largest reported BET to date. Figure 2 illustrates that the optimal conditions for the synthesis of HKUST-1 yielded large crystals, while the samples with a lower BET

showed intergrowth and other deviations that are not captured by powder diffraction analysis. Since the BET of 2045 $m^2 g^{-1}$ close to the theoretical maximum of 2153 $m^2 g^{-1}$[22], there was no need to further continue our GA using the BET as objective function.

**Capturing chemical intuition using machine learning.** The common practice is to claim victory and publish the synthesis conditions that yielded the highest experimentally measured BET value. Instead, we would like to focus on the observation that to achieve this high BET surface area, we have over 120 failed and partly successful experiments. In the following, we analyze this data to quantify the relative importance of the experimental variables on the outcome of the synthesis. We use the embedded technique in random decision forest, a machine learning regression model (See Supplementary Note 8 for discussion and comparison to other techniques). The result is shown in Fig. 3a and provides the relative impact of the probed experimental parameters on crystallinity and phase purity. For example, changing the temperature has three times more impact than changes in the reactant ratio. It is this type of information that a synthetic chemist will typically transfer to the next experiments; knowingly, as rules of thumb, or, subconsciously, in the form of "chemical intuition." Machine learning of the recorded data allows us to quantify this intuition, and to use it for subsequent experiments.

Without prior knowledge, the difference between synthesis conditions was quantified as the Euclidean distance in 9D space using an equal weight of all parameters. Building on the chemical intuition extracted from our machine-learned model, we now compute the distance in 9D space using the chemical intuition to weight each dimension in the distance measure. If we normalize these weights such that the most important variable has a value of

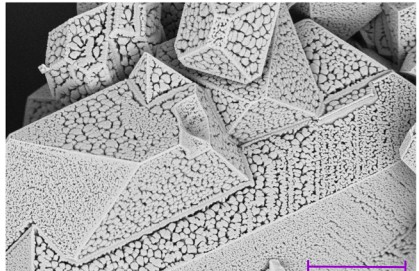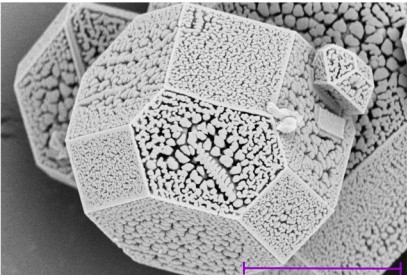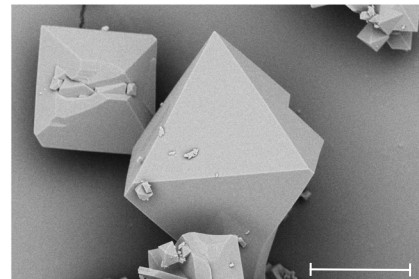

**Fig. 2** Scanning electron micrograph of several Cu-HKUST-1 samples. All these samples have high crystallinity (See Supplementary Note 10) but show a wide range of surface areas (see Table 1 for surface areas and Supplementary Figure 11 for more images). Scale bars for sample 1, sample 3 and sample 5 show 5 µm, 4 µm, and 10 µm, respectively. (HKUST = Hong Kong University of Science and Technology)

1, we obtain a chemical space shown in Fig. 3b. This figure shows how the chemical space for HKUST-1 shrinks in the new metric (the Euclidian distance, weighted by the importance of variables), illustrating that less samples can be placed along less important dimensions without loss of sampling accuracy. Therefore, since the chemical space can be sampled much more efficiently, the chance of success is larger for the same number of trials.

**Application of learned chemical intuition**. We now illustrate transferring the quantified chemical intuition to a new synthesis. Most studies on HKUST-1 are focused on the Cu(II) version, but HKUST-1 can also be synthesized with Zn(II)[23]. We can now take three approaches to synthesize Zn-HKUST-1: First, we could assume that the synthesis of Zn-HKUST-1 to be similar to Cu-HKUST-1 and simply reuse the successful conditions of Cu-HKUST-1. For our case, the equivalent of a literature search of successful synthesis conditions for Cu-HKUST-1 is simply testing those optimal synthesis conditions we found for Cu(II). None of the top ten synthesis conditions for Cu(II) yield crystals for Zn(II). Without chemical intuition, this would put us back to square one, and we would have to restart the procedure, i.e., we use the same set of most diverse conditions as used for Cu-HKUST-1. Using our chemical intuition, however, we can sample the space more intelligently by assigning the previously determined importance of variables, resulting in denser sampling of more important experimental parameters. For this weighted set of 20 diverse conditions, two conditions yielded Zn-HKUST-1 crystals.

The difference in weighted and unweighted synthesis conditions is illustrated in Fig. 4. As we are sampling a high-dimensional space with a low number of points, the most diverse conditions lie at the boundaries of each dimension, and only start populating the interior with sufficient sample points. In the weighted space representation (Fig. 4b), the set generated without prior knowledge includes several points that are so close to each other that they are not expected to yield additional information. Having determined the (lack of) variation of the sample fitness for the different variables, the variables of lesser importance may be sampled less frequently without loss of accuracy. In fact, the reweighted set samples the most important parameters roughly 10 times more frequently than the least important ones.

We note that our 20 intuition-based samples would need to be replaced by order of four to five thousand samples without intuition in order to maintain the same sampling accuracy (see Supplementary Note 7), illustrating a dramatically increased chance of successful synthesis for a chemist who leverages chemical intuition.

Th example of Cu-HKUST-1 and Zn-HKUST-1 illustrate how quantifying and reusing chemical intuition can be beneficial in a case, where the chemistry is too specific for the synthesis conditions themselves to be transferable. In this work, we selected HKUST-1 as a case study to illustrate the methodology.

**Discussion**

The main aim of this work was to develop a simple, yet powerful framework that allows one to use failed and partially successful experiments to systematically improve synthesis strategies. This framework does not rely on a detailed understanding of how the different synthesis conditions impact the outcome. Rather, it relies on the notion that, over the course of many experiments, chemists develop an intuition on how to approach the problem of finding the right synthesis conditions. Here, we have developed a simple way of capturing this chemical intuition using machine learning.

Our case study of HKUST-1 was intended as a proof of principle that we can capture and quantify chemical intuition, and effectively use it to develop more efficient synthesis strategies. We note that the data produced in this work are ideal from a machine learning point of view. Using a robotic platform provides precise control over the synthesis variables which results in less noise in the outcome of reactions and improved reproducibility (See Supplementary Note 12 for details). Furthermore, we are using only one synthesis technique. This allows obtaining an accurate estimate of the chemical intuition using a relatively small set of experiments. If all groups that have worked on the synthesis of HKUST-1 would have published also their failed and partially successful experiments, the data would be significantly less homogenous because of other influencing variables, e.g., size of reactor, purity of reactants, etc., but the much larger data set would also make it easier for machine learning to filter out these inhomogeneities.

Figure 5 summarizes how we envision the three components of our framework, synthesis, optimization, and machine learning, to interact. For example, one can use the GAs to optimize the synthesis conditions while, in parallel, machine learn the relative importance of the experimental variables, leading to more rational experiments. This is the approach we have used for HKUST-1. For more complex synthesis, however, one can take this approach one step further by leveraging the machine learning model in a second way: to score the next generations of the genetic algorithm in silico, going back to experiment only once convergence is reached. Appropriately fine-tuned, this has the potential to significantly reduce the number of experiments required (See Supplementary Note 4 for details).

An important practical question is how we envision our approach can be used by other groups. The screening strategy we used can be easily adapted to other synthesis problems. Define the chemical space, generate the most diverse set of conditions, and use a combination of GAs and machine learning to find the optimal target. Of course, one can only take advantage of the

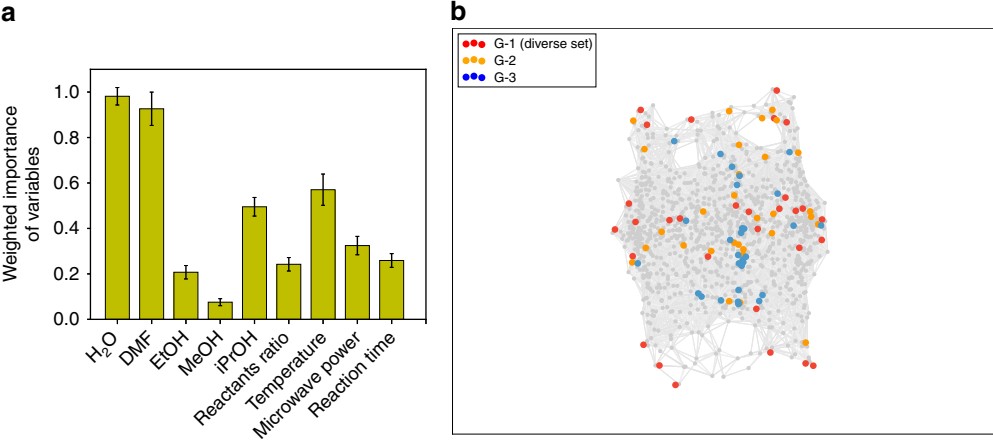

**Fig. 3** Captured chemical intuition and the chemical space in the new metric. **a** Relative impact of the 9 parameters on Cu-HKUST-1 synthesis, as obtained from machine learning. Maximum impact is normalized to one. The error bars show the standard deviation of the relative importance of variables over 1000 retraining of the random forest with different unique random seeds. **b** Multidimensional scaling projection of the experimental conditions, in which the distance is weighted by the relative importance of the variables. The colour of dots indicates the generation in the GA. The grey dots represent the chemical space in the new metric. Grey dots are the 1000 most diverse conditions obtained using MaxMin method without weighting distances. (HKUST = Hong Kong University of Science and Technology)

"chemical intuition" in generating the set of most diverse conditions if we have a sufficient number of failed or partially successful experiments using a similar synthesis technique and similar chemical space. A key component here is that the more groups share their failed and partially successful experiments, the more versatile the model's chemical intuition will become. In this respect, each MOF synthesis group has a similar challenge, once the ligands and metal nodes are synthesized: how to find the right synthesis conditions that crystals will form? The quantified "intuition" by machine learning is by no means different from the intuition developed by chemist in the lab; it is useful in many cases, but one always need to keep in mind that in some cases the chemistry can be surprisingly different. The software we have developed for this study is available as a web application on the Materials Cloud[24], together with the "chemical intuition" which we will be continuously updating and adopting to the needs of the community. If a large number of groups involved in MOF synthesis agree on a systematic reporting of failed or partially successful experiments, this can be an extremely powerful tool that has the potential to change the way our research community approach synthetic chemistry.

## Methods

### Methodology overview
To reconstruct the not reported (partially) failed and successful data in the literature, we simulate the steps that are taken by someone with no chemical intuition for synthesis of a MOF by a genetic algorithm (GA) optimization procedure. We start with the set of most diverse synthesis conditions based on a simple algorithm for the MaxMin diversity problem[20]. Chemical intuition can be incorporated by assigning appropriate weights to different variables. The diverse set constitutes the first generation of the optimization cycle. A robotic synthesis and characterization approach is used for synthesis of MOFs, and measurement of X-ray diffraction patterns. We rank the experiments based on their crystallinity and BET surface area. This ranking is fed to the genetic algorithm to generate a new generation of synthesis conditions. Afterwards, the new generation is synthesized and characterized. This procedure continues until it satisfies the objective function of the synthesis. All the data generated in the synthesis procedure is used to train a machine learning model to assess the importance of synthesis variables. Below we summarize the main steps for each part of this procedure. A more detailed description can be found in the Supporting Information.

### Genetic algorithm
The genetic algorithm (GA) was used as it is implemented in the global optimization toolbox of MATLAB[25]. The population of each generation was fixed to thirty. At each step, the GA was initialized with the last generation and

used its individuals' fitness. Migration, crossover and mutation genetic functions were applied. The ranking of the individuals was used as the fitness function which determines the chance of each parent in generating children in new generation. The optimization starts with the set of most diverse individuals (see Supplementary Note 2) to ensure exploration of the chemical space with no bias. For details, see Supplementary Note 1.

### Robotic synthesis and characterization
The synthesis was carried out in a microwave synthesis reactor (Biotage, Uppsala, Sweden) affixed on a HT robotic platform (Chemspeed technologies, Füllinsdorf, Basel, Switzerland). The synthesis steps inclusive of handling and dispensing of the reactants (metal salt, ligand, solvents) in to the microwave reaction vials, stirring of the dispensed reactant mixture, capping, crimping, and the transportation of the microwave reaction vials to the microwave reactor cavity was completely automated and executed using the Chemspeed autosuite software. All the chemicals were purchased from commercial sources and used without further purification.

Powder X-ray diffraction (PXRD) patterns were collected using the powder diffractometer Bruker D8 Advance with TWIN/TWIN optics and LYNXEYE XE-T detector equipped with high throughput sample changer. The samples were loaded on a silicon (no background) sample holder and the PXRD pattern was collected in a 2θ range between 2–20 using a monochromatic copper (Cu) X-ray source (λ = 1.54056 Å). The sample holders were rotated about their central axis during data collection, minimizing potential effects from preferred orientation. The diffractometer was controlled using the Bruker's EVA software. All measurements were performed at room temperature. Crystallinity and phase purity of samples were assessed by the full-width at half maximum (FWHM) of the diffraction peaks of the samples' powder X-ray diffraction patterns, and with a penalty in fitness for extra peaks compared to the simulated pattern. $N_2$ isotherms (77 K) were recorded to apply the Brunauer–Emmett–Teller (BET) model in the relative pressure range of 0.05–0.30 to determine the surface area of the HKUST-1 MOFs. The isotherms were collected by using an IGA system (Intelligent Gravimetric Analyzer, Hiden Isochema Ltd., Warrington, UK) and the BELSORP mini system (MicrotracBEL Corp., Osaka, Japan). Prior to isotherm collection, the HKUST-1 samples were activated at 220 °C under dynamic vacuum for 6 h to get the desolvated HKUST-1 (dark blue).

### Machine learning
The random forest ensemble learner was used for assessing the importance of variables[26]. Random forest is a supervised learning algorithm for classification and regression problems. A bootstrapped aggregated forest of 200 decision trees with maximum depth of three was trained to predict the outcome of the synthesis based on the synthesis variables. The mean absolute error (MAE) of the predictions was smaller than 9% and 14% for cross-validation and not seen data points, respectively. The importance of variables was estimated by permuting out-of-bag observations. The machine learning algorithm was implemented first using the statistics and machine learning toolbox of MATLAB, and then ported to python (using the scikit-learn package[27]) for the web application. For more details see Supplementary Note 3.

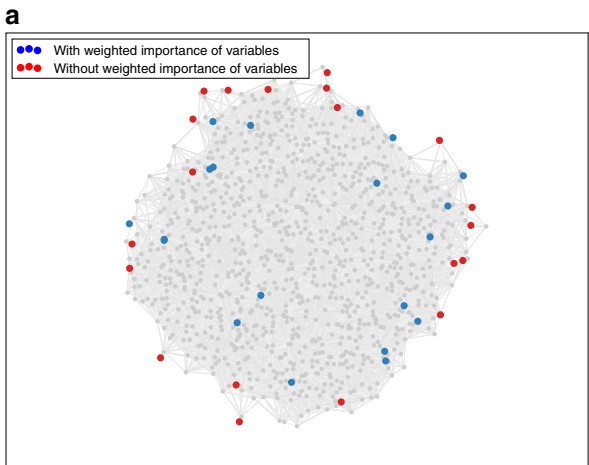
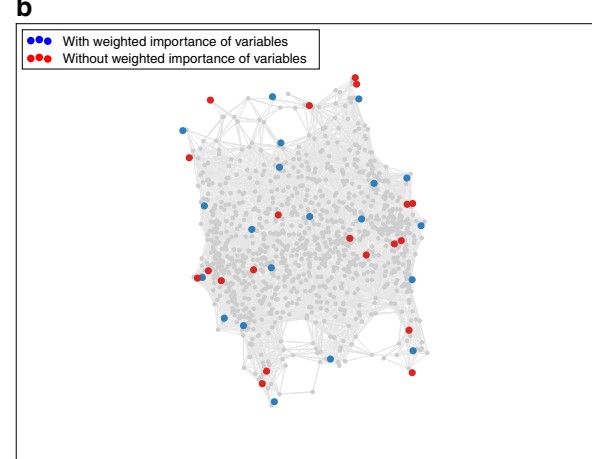

**Fig. 4** Distribution of diverse sets in the chemical space of Zn-HKUST-1. Multidimensional scaling projection of the set of 20 most diverse synthesis conditions with (blue) and without (red) taking the relative impact of synthesis variables into account. Both sets are shown in the unweighted space (**a**) and in the weighted space (**b**). (HKUST = Hong Kong University of Science and Technology)

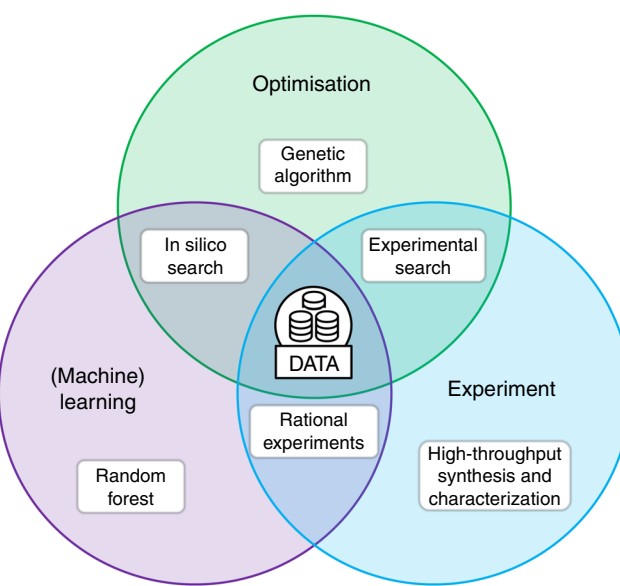

**Fig. 5** Schematic of the components of the framework used for MOF synthesis

**Multidimensional scaling plots**. Multidimensional scaling (MDS) provides a visual representation of data based on the pairwise distances, similarity or dissimilarity within a set of points in a high-dimensional space. Here, we choose metric MDS using the weighted Euclidean pairwise distances between points in both high-dimensional (HD) and low-dimensional (LD) spaces. The algorithm aims to preserve the HD distances between objects in the LD representation. The metric for evaluation of how accurate the LD representation is compared to the high-dimensional distances is called the stress function:

$$S = \left( \sum_{i,j=1,...N} d_{i,j} - \bar{d}_{i,j} \right)^{1/2}.$$

This function returns the residual sum of squares of the distances in the HD space ($d$) to the LD space ($\bar{d}$). We use stress majorization algorithm to minimize the stress function as implemented in scikit-learn python package. The weights in the weighted Euclidian distance function, $d_{a,b} = \sqrt{\sum_i^n w_i (a_i - b_i)^2}$, are set to 1 for all variables in Figs. 1a and 4a (no chemical intuition), and equal to the weighted importance of variables in Figs. 3b and 4b (using chemical intuition).

**Code availability**. The developed web application software for this study is available in Materials Cloud via https://doi.org/10.24435/materialscloud:2018.0011/v3.

**Data availability**
All the data and the developed software in this manuscript are available in Materials Cloud via https://doi.org/10.24435/materialscloud:2018.0011/v3. Access to any other materials can be requested by writing to the corresponding author.

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

## Acknowledgements

This research was supported by the NCCR MARVEL, funded by the Swiss National Science Foundation. S.M.M. was supported by the Deutsche Forschungsgemeinschaft (DFG, priority program SPP 1570). K.C.S. was supported by the Swiss National Science Foundation (SNSF) with funding under the Ambizione Energy Grant n. PZENP2_166888. MH acknowledges support from the Spanish Ministry of Economy and Competitiveness (RYC-2013-13949). B.S. research has received funding from the European Research Council (ERC) under the European Union's Horizon 2020 research and innovation programme (grant agreement No 666983, MaGic). The live version of web application is supported by cloud resources provided by the Swiss National Super-
computing Centre (CSCS). S.M.M. thanks Dr. Peter G. Boyd and Dr. Pascal Schouwink for useful discussions.

## Author contribution

S.M.M., L.T., M.H., and B.S. performed the computational part, and A.C. and K.C.S. performed the experimental part of the work. S.M.M. and L.T. prepared the web application. All authors contributed in design of the project, interpretation of the results and writing the manuscript.

## Additional information

**Competing interests:** The authors declare no competing interests.

