## [Peer Review File · Nature Communications]

Reviewers' comments:

Reviewer #1 (Remarks to the Author):

The authors of the paper describe the use of machine learning to efficiently find optimize synthesis conditions for MOFs. At least, that is the overall vision, the paper itself is more of a prototype / proof-of-concept to show that maybe it can be done.

First, let me begin with the positive comments, which is to say that I absolutely love the idea and it was a very engaging paper (although a bit too repetitive, see my comments below). Marrying machine learning and robotic synthesis is **clearly** the way of the future, not only for MOFs, but for chemical synthesis in general, and it is fantastic to see these authors pioneer that promising frontier.

That being said, there are several areas where the paper could be significantly improved. The most major concern I have with the paper is that the authors do not really show that machine learning is beneficial relative to much simpler analysis of the data. It's not clear that it would not have worked equally well to generate 120 data points (equidistant in the chemical space) and then use principle component analysis to identify the most important factors (that would then form the basis of a rule of thumb). My gut feeling is that the authors are right and that machine learning is the way to go, but the paper doesn't demonstrate this and it is arguably the main claim of the paper.

I have listed other points below roughly in order as they appear in the paper:

- I frequently read (and presumably other readers of Nature Communications have too) that machine learning requires very large datasets to be effective. Is 120 really enough? The authors should comment on the size of the dataset and perhaps be upfront about the limitations of their methodology.
- The paper would benefit from another editing pass, particularly to remove the frequent repetition of the phrase "machine learning allows us to quantify this intuition." Variations of that sentence appear in almost every paragraph of the paper.
- In all of the projection figures, such as Figure 1a and Figure 3b, the visualization technique is providing very little useful information to the reader. There are tick marks, for example, but no numbers or axes labels. It's very unclear what is being shown. For example, if we started with nine dimensions and went down to two, which are the two remaining that are being shown?
- In Figure 1b, it appears that the maximum crystallinity decreases in Gen 3. Why is this? Shouldn't the best parent carry forward in the GA search (i.e., elitism)? The SI isn't very precise, I couldn't find the exact fitness function used (I only know it has something to do with the crystallinity ranking). Also the equation for the crossover scheme appears to have a hyphenated "random number" word which appears as "random - number" which is confusing. Perhaps it is intended to be a minus sign, in which case I do not understand the meaning of the equation.
- On pg. 5 the authors write "This figure shows how the chemical space for HKUST-1 shrinks in the new metric..." This is totally unclear. Perhaps it is obvious to the authors but I cannot see the

"new metric" in Figure 3b and hence it is very hard to see how the chemical space "shrinks." Is there a meaningful definition of volume here for which the word "shrink" applies? What does "along the new metric" mean with respect to Figure 3b? Does the new metric correspond to one of the axes in the 2D projection?

- Figure 4 is similarly not intuitive. Also the figure legend refers to "variable importance" which is only mentioned in passing in the text (presumably that is another way of saying "weighted importance" but for a confused reader it is helpful to use the exact same words whenever possible).

- In addition to the plots shown, I think it would be helpful (and easy to do!) to include a figure (or some variation of it), that shows crystallinity/fitness on the y-axis and one of the synthesis variables on the x-axis (e.g., H₂O, temperature) and show the whole 120 point dataset. Please do this keeping the same axes but showing Cu-HKUST and Zn-HKUST next to each other (I would avoid overlapping the data points as that would get messy). That's the unsophisticated non-machine learning way that humans extract intuition from data, and it would really complement the other figures.

- There's one claim made twice in the paper that readers may doubt, namely that the robotic synthesis ensures perfect reproducibility. Do we know that this is true? More importantly, do you show evidence for this? One might expect that even when the synthesis conditions are 100% identical, the resulting crystallinity/surface area might vary considerably. It would be helpful for the authors to convey how much variability there is when trying to reproduce a synthesis.

- The first paragraph of the "Methods" section is an almost verbatim repetition of the method described in the beginning of the paper. Unless the format of the journal requires this level of redundancy, I would suggest removing it.

As a final comment, I would like to reiterate that I think this is amazing research. This manuscript needs a bit more work but I think it will be really highly cited once it is out there.

Reviewer #2 (Remarks to the Author):

Excellent article by Berend and co-workers on Capturing chemical intuition in synthesis of metal-organic frameworks. Using machine learning algorithms authors try to come with possible synthetic parameters to produce MOFs with high crystallinity and surface area. Given the magnitude of experimental parameters that we could manipulate during MOF synthesis, I believe machine learning could be used a tool to predict accurate MOF synthetic parameters. However I have couple of comments...

1) What is the role of activation on all the synthesized HKUST-1 MOFs from the this database?. If this the activation play a huge role in improving crystallinity and surface area, i believe the chemical composition and temperature does not have a major role.

2) I recommend to provide all the PXRD patterns of at different conditions and compare with top 5 crystalline HKUST-1

3) Apart from the temperature, i believe mixing plays a huge role in synthesis. The constant stirring reduces the cold spots in the reaction (at least in the bulk synthesis)

Reviewer #3

The authors report on a set of computer programs that will support the evaluation of complex parameter space and thus the discovery and synthesis optimization of a new compound. This

methodology was used to optimize the synthesis of the well-known HKUST-1 and the corresponding known compound containing Zn²⁺ ions. The paper has a much higher impact if a new material would have been discovered and optimized. The methodology uses reported synthesis procedures as well as results obtained employing a genetic algorithm (GA) optimization procedure.

The properties (objective function) of the compounds are determined and the relevance of synthetic parameters is identified and quantified. The GA procedure is employed - ideally using a robotic system - until the objective function is accomplished. This procedure is "similar" to the one used every day by the synthetic chemist and the authors want to capture the experience of the chemist, which they call chemical intuition, using machine learning. The advantage of machine learning definitely is the identification of non-obvious parameters.

Overall, the manuscript could be very interesting to the synthetically working chemist if the following points are taken into account

- 1) The results of the characterization cannot be found in the manuscript or the SI. What are the composition of, at least, the optimized materials? A thorough characterization (elemental analysis, thermogravimetric measurements, IR spectroscopy ...) needs to be carried out since, for example, structural defects are often observed in MOFs, which lead to improved specific surface areas.
- 2) Sorption properties of certain MOFs and especially of HKUST-1, are known to be highly dependent on thermal and chemical treatment. How were the samples treated, perhaps the treatment was different compared to the one by other groups and this is the decisive parameter? Evaluation of the sorption isotherms using simply the BET model and the relative pressure range 0.05 to 0.3 is not sufficient. The community has agreed on using the method reported by Rouquerol et al.
- 3) Transferring chemical parameters from Cu²⁺ to Zn²⁺ chemistry is very "bold". The chemistry of the metal ions is quite different and I propose that the authors were just lucky to find in the diverse set of variables one hit. Thus capturing of chemical intuition by machine learning for a given chemical system is really good, but the transferability is very limited.
- 4) I fully agree with the authors that chemists need to report also the "failed" experiments, i.e. the ones that did not lead to the desired product. This gives valuable information for synthetic chemists trying to reproduce the results in another laboratory. But a very important problem, that is not covered in the manuscript, is the fact that the many more variables are influencing the properties of the product. Especially the size of the reactor employed but also things like purity of the starting materials or solvents is very important.

minor points

- 1) The reactions were not carried out in parallel (but serial), this is not possible with the microwave oven employed in the study.
- 2) Fig. S6: please correct "Powder x-rad diffraction"
- 3) "Machine learning: capturing chemical intuition" The first sentence is not true any more. Nowadays MOF scientists are more interested in other topics such as the properties, the up-scaling or

the how the MOFs are formed. The search for “world- record” BET is from yesterday. The expression “world record BET” does not make any sense (BET-value, BET theory, ...).

4) Figure 2: scale bars are missing/not readable.

Response to Reviewer #1

The authors of the paper describe the use of machine learning to efficiently find optimize synthesis conditions for MOFs. At least, that is the overall vision, the paper itself is more of a prototype / proof-of-concept to show that maybe it can be done.

First, let me begin with the positive comments, which is to say that I absolutely love the idea and it was a very engaging paper (although a bit too repetitive, see my comments below). Marrying machine learning and robotic synthesis is *clearly* the way of the future, not only for MOFs, but for chemical synthesis in general, and it is fantastic to see these authors pioneer that promising frontier.

That being said, there are several areas where the paper could be significantly improved. The most major concern I have with the paper is that the authors do not really show that machine learning is beneficial relative to much simpler analysis of the data. It's not clear that it would not have worked equally well to generate 120 data points (equidistant in the chemical space) and then use principle component analysis to identify the most important factors (that would then form the basis of a rule of thumb). My gut feeling is that the authors are right and that machine learning is the way to go, but the paper doesn't demonstrate this and it is arguably the main claim of the paper.

Authors reply:

We want to point out that one of the central aims of this study is to initiate and introduce usage of machine learning (ML) in MOF synthesis. We developed and provide tools that we hope will encourage MOF synthesis groups to store/report their data systematically to be used to train a powerful ML predictor of synthesis outcome for a wide range of MOF chemistries. We agree with the reviewer that the analysis of importance of variables can be done with other simpler techniques despite their limitations (see below), however, our choice of methodology was highly motivated with the fact that this study opens a great opportunity for following studies in MOF synthesis.

Somewhat hidden in the manuscript we have mentioned that the ML is not only a convenient method to identify the most important factors, but it can also be used to predict the next generation for genetic algorithms, which is an important advantage for more complex synthesis procedures. We have now mentioned this more prominently:

“For more complex synthesis, however, one can take this approach one step further by leveraging the machine learning model in a second way: to score the next generations of the genetic algorithm in silico, going back to experiment only once convergence is reached. Appropriately fine-tuned, this has the potential to significantly reduce the number of experiments required (See SI for details).”

In addition, in the revised manuscript, we have added discussion on using other simpler techniques for feature selection and importance of variables in the main text:

“In the following, we analyze this data to quantify the relative importance of the experimental variables on the outcome of synthesis. Here we use the embedded technique in random decision forest, a machine learning regression model (See SI for discussion and comparison to other techniques).”

The discussion can be found in SI section “Importance of variables: other techniques”, where we compare ML with two statistical filters.

I have listed other points below roughly in order as they appear in the paper:

- I frequently read (and presumably other readers of Nature Communications have too) that machine

learning requires very large datasets to be effective. Is 120 really enough? The authors should comment on the size of the dataset and perhaps be upfront about the limitations of their methodology.

Authors reply:

Although machine learning has become more popular after engaging with big data, it is indeed a common misconception that machine learning can only be used on large volume of data. Examples of application of machine learning for small data sets includes the analysis of patients' genes for cancer diagnosis (typically smaller than 100 data points), the Iris flower dataset (150 data points) which is extensively used for testing machine learning models and the Berkeley Lab's "Minimalist Machine Learning" recently developed algorithms for analyzing images from very little data.¹ In all these cases, the number of training data are in order of our dataset. However, for small datasets, one need to be more cautious in selecting the model, training and overfitting.

- The paper would benefit from another editing pass, particularly to remove the frequent repetition of the phrase "machine learning allows us to quantify this intuition." Variations of that sentence appear in almost every paragraph of the paper.

Authors reply:

In the revised manuscript we did our best to avoid repetition.

- In all of the projection figures, such as Figure 1a and Figure 3b, the visualization technique is providing very little useful information to the reader. There are tick marks, for example, but no numbers or axes labels. It's very unclear what is being shown. For example, if we started with nine dimensions and went down to two, which are the two remaining that are being shown?

Authors reply:

We believe we were not sufficiently clear on the multidimensional scaling (MDS) plots and their interpretation. We added a description of what is shown in the main text:

"Figure 1(a) shows these conditions in a multidimensional scaling (MDS) projection. MDS-plots visualize the similarity between individuals in a dataset.²¹ In this study, the Euclidian distance of normalized variables measures the similarity between synthesis trials. In Figure 1(a), similar synthesis trials are mapped close to each other while dissimilar experiments are far from each other on the map (see method section for details)."

Moreover, we moved the description of MDS plots from SI to the method section for clarity:

"Multidimensional scaling plots. Multidimensional scaling (MDS) provides a visual representation of data based on the pairwise distances, similarity or dissimilarity within a set of points in a high-dimensional space. Here, we choose metric MDS using the weighted Euclidean pairwise distances between points in both high-dimensional (HD) and low-dimensional (LD) spaces. The algorithm aims to preserve the HD distances between objects in the LD representation. The metric for evaluation of how accurate the LD representation is compared to the high-dimensional distances is called the stress function: $S = (\sum_{i,j=1..N} d_{i,j} - \bar{d}_{i,j})^{1/2}$.

This function returns the residual sum of squares of the distances in the HD space (d) to the LD space (\bar{d}). We use stress majorization algorithm to minimize the stress function as implemented in scikit-learn python package. The weights in the weighted Euclidian distance function, $d_{a,b} = \sqrt{\sum_i^n w_i (a_i - b_i)^2}$, are set to 1

¹ Pelt, Daniël M., and James A. Sethian. "A mixed-scale dense convolutional neural network for image analysis." *Proceedings of the National Academy of Sciences* 115.2 (2018): 254-259.

<https://newscenter.lbl.gov/2018/02/21/new-berkeley-lab-algorithms-create-minimalist-machine-learning-that-analyzes-images-from-very-little-information/>

for all variables in figures 1(a) and 4(a) (no chemical intuition), and equal to the weighted importance of variables in figure 3(b) and 4(b) (using chemical intuition)."

- In Figure 1b, it appears that the maximum crystallinity decreases in Gen 3. Why is this? Shouldn't the best parent carry forward in the GA search (i.e., elitism)? The SI isn't very precise, I couldn't find the exact fitness function used (I only know it has something to do with the crystallinity ranking). Also the equation for the crossover scheme appears to have a hyphenated "random number" word which appears as "random - number" which is confusing. Perhaps it is intended to be a minus sign, in which case I do not understand the meaning of the equation.

Authors reply:

For the third generation, we rank the samples based on both their BET and crystallinity. Therefore, the sample with the highest crystallinity is not necessarily the best sample. The hyphen in the crossover function was a typo and removed in the revised version.

To explain better the fitness function, we added more explanation in SI:

"To generate a new generation, the GA takes the chromosomes of the past generation and their corresponding fitness score based on an objective function. The fitness functions in current study are crystallinity or crystallinity and BET surface area of the first and the second generations, respectively. We use the full width at half maximum of powder x-ray diffraction patterns as a measure of crystallinity of samples (see below)."

Moreover, the equation used to estimate FWHM is stated in SI:

"The crystallinity of each sample was assessed by the full width at half maximum (FWHM) of powder X-ray diffraction patterns (PXRD) (3–5). We start with separating peaks in the PXRD. Afterwards, a Gaussian function is fitted to the peak which give us the FWHM with the following set of equations:

$$f(x) = \frac{1}{\sigma\sqrt{2\pi}} \exp \left[-\left(\frac{(x - x_0)^2}{2\sigma^2} \right) \right], \quad (1)$$

$$FWHM = 2\sqrt{2\ln(2)}\sigma, \quad (2)$$

where variable x is the 2θ of the diffraction angle. The average FWHM of all the peaks of the PXRD is taken as the measure of crystallinity. Lorentzian, Pearson, and combined Lorentzian, Pearson and Gaussian distributions were also considered, and no considerable differences were observed in the ranking."

- On pg. 5 the authors write "This figure shows how the chemical space for HKUST-1 shrinks in the new metric..." This is totally unclear. Perhaps it is obvious to the authors but I cannot see the "new metric" in Figure 3b and hence it is very hard to see how the chemical space "shrinks." Is there a meaningful definition of volume here for which the word "shrink" applies? What does "along the new metric" mean with respect to Figure 3b? Does the new metric correspond to one of the axes in the 2D projection?

Authors reply:

Look the explanation above about MDS plots. Additionally, we added the meaning of the "new metric" at the place it is referred to:

"This figure shows how the chemical space for HKUST-1 shrinks in the new metric (the Euclidian distance, weighted by the importance of variables), illustrating that less samples can be placed along less important dimensions without loss of sampling accuracy."

- Figure 4 is similarly not intuitive. Also the figure legend refers to "variable importance" which is only mentioned in passing in the text (presumably that is another way of saying "weighted importance" but for a confused reader it is helpful to use the exact same words whenever possible).

Authors reply:

The plots are adopted accordingly in the revised manuscript.

- In addition to the plots shown, I think it would be helpful (and easy to do!) to include a figure (or some variation of it), that shows crystallinity/fitness on the y-axis and one of the synthesis variables on the x-axis (e.g., H₂O, temperature) and show the whole 120 point dataset. Please do this keeping the same axes but showing Cu-HKUST and Zn-HKUST next to each other (I would avoid overlapping the data points as that would get messy). That's the unsophisticated non-machine learning way that humans extract intuition from data, and it would really complement the other figures.

Authors reply:

We plotted crystallinity with respect to the 9 variables in our synthesis and added to the SI.

- There's one claim made twice in the paper that readers may doubt, namely that the robotic synthesis ensures perfect reproducibility. Do we know that this is true? More importantly, do you show evidence for this? One might expect that even when the synthesis conditions are 100% identical, the resulting crystallinity/surface area might vary considerably. It would be helpful for the authors to convey how much variability there is when trying to reproduce a synthesis.

Authors reply:

Of course, the reviewer is right. The robot is not infinitely precise but works within the specifications. In the revised manuscript we applied modifications in the main text:

"Using a robotic synthesis platform improves the reproducibility of the generated data.",
and,

"We note that the data produced in this work are ideal from a machine learning point of view. Using a robotic platform provides precise control over the synthesis variables which results in less noise in the outcome of reactions and improved reproducibility (See SI for details)."

Furthermore, we conducted a series of "identical" experiments to illustrate the reproducibility of outcomes and how this impact the outcomes of synthesis. The full analysis can be found in SI section "Accuracy and reproducibility of experiments". This section includes the synthesis steps of the robotic synthesis, the precision of robot in controlling the synthesis variables and the impact of this on the BET surface area and the PXRD of samples. We report the average BET surface areas in the main text in the revised version.

- The first paragraph of the "Methods" section is an almost verbatim repetition of the method described in the beginning of the paper. Unless the format of the journal requires this level of redundancy, I would suggest removing it.

Authors reply:

Although it might sound repetitive, we rather keep this paragraph because it helps the readers to easily grasp the steps of our methodology.

As a final comment, I would like to reiterate that I think this is amazing research. This manuscript needs a bit more work but I think it will be really highly cited once it is out there.

Response to Reviewer #2

Excellent article by Berend and co-workers on Capturing chemical intuition in synthesis of metal-organic frameworks. Using machine learning algorithms authors try to come with possible synthetic parameters to produce MOFs with high crystallinity and surface area. Given the magnitude of experimental parameters that we could manipulate during MOF synthesis, I believe machine learning could be used a tool to predict accurate MOF synthetic parameters. However I have couple of comments...

1) What is the role of activation on all the synthesized HKUST-1 MOFs from the this database? If this the activation play a huge role in improving crystallinity and surface area, i believe the chemical composition and temperature does not have a major role.

Authors reply:

We fully agree with the reviewer that the activation plays a crucial role on the properties of the crystals, and therefore, we activated all the samples included in this work using the same conditions of 220°C for 6 hours under vacuum. In addition, in the revised manuscript, we show that different synthesis techniques also could yield phase pure crystalline HKUST-1 with the wide range of BET surface areas despite the same activation (See reply to reviewer 3 for more details).

2) I recommend to provide all the PXRD patterns of at different conditions and compare with top 5 crystalline HKUST-1

Authors reply:

In the revised version, all the PXRDs of the samples included in the SI with their fitness scores for comparison with the optimal samples.

3) Apart from the temperature, i believe mixing plays a huge role in synthesis. The constant stirring reduces the cold spots in the reaction (at least in the bulk synthesis)

Authors reply:

We agree with the reviewer on the importance of stirring in bulk synthesis. In this context, for all the synthetic reactions performed in this work, the reaction solution was stirred on the robotic platform for 5 minutes before it was subjected to microwave irradiation. Moreover, we take advantage of using microwave chemistry with fast and homogenous heating of the reactants to avoid cold spots in reactions. In the revised manuscript, we have included the full program of robotic synthesis in the SI section “Accuracy and reproducibility of experiments” (Supplementary Figure 18).

Response to Reviewer #3

The authors report on a set of computer programs that will support the evaluation of complex parameter space and thus the discovery and synthesis optimization of a new compound. This methodology was used to optimize the synthesis of the well-known HKUST-1 and the corresponding known compound containing Zn²⁺ ions. The paper has a much higher impact if a new material would have been discovered and optimized.

The methodology uses reported synthesis procedures as well as results obtained employing a genetic algorithm (GA) optimization procedure. The properties (objective function) of the compounds are determined and the relevance of synthetic parameters is identified and quantified. The GA procedure is employed - ideally using a robotic system - until the objective function is accomplished. This procedure is “similar” to the one used every day by the synthetic chemist and the authors want to capture the experience of the chemist, which they call chemical intuition, using machine learning. The advantage of machine learning definitely is the identification of non-obvious parameters.

Overall, the manuscript could be very interesting to the synthetically working chemist if the following points are taken into account

- 1) The results of the characterization cannot be found in the manuscript or the SI. What are the composition of, at least, the optimized materials? A thorough characterization (elemental analysis, thermogravimetric measurements, IR spectroscopy ...) needs to be carried out since, for example, structural defects are often observed in MOFs, which lead to improved specific surface areas.

Authors reply:

In addition to the SEM, PXRD and BET characterization we have included the results of the elemental analysis, thermogravimetric analysis and IR spectra of the optimal HKUST-1 MOFs to the SI section “Supplementary information: Cu-HKUST-1 samples”.

- 2) Sorption properties of certain MOFs and especially of HKUST-1, are known to be highly dependent on thermal and chemical treatment. How were the samples treated, perhaps the treatment was different compared to the one by other groups and this is the decisive parameter? Evaluation of the sorption isotherms using simply the BET model and the relative pressure range 0.05 to 0.3 is not sufficient. The community has agreed on using the method reported by Rouquerol et al.

Authors reply:

We truly agree with the reviewer on the crucial role of thermal and chemical treatment on the sorption properties of HKUST-1. However, we remind that all the samples included in this work were activated at the same conditions of 220°C for 6 hours under vacuum.

Furthermore, to illustrate that the different synthesis techniques used by other groups could also yield phase pure crystalline HKUST-1 with the wide range of BET surface areas despite the same activation procedure, we carried out a series of experiments synthesizing HKUST-1 through different synthetic routes, namely ultrasonication (US), reflux, conventional electric heating (CEH) with oven and compared it with the HKUST-1 synthesized with the aid of microwave. The PXRD patterns of the as-made samples were investigated and they are in concordance with the simulated HKUST-1 pattern as shown in the Supplementary Figure 7.

However, the sorption studies indicated that the HKUST-1 MOFs possess different BET surface areas though they were all activated under the same conditions of 220°C for 6 hours under vacuum. The BET surface areas were determined to be as follows: 389 m² g⁻¹ for CE HKUST-1; 1228 m² g⁻¹ for MW HKUST-1, 1036 m² g⁻¹ for US HKUST-1, 900 m² g⁻¹ for reflux HKUST-1.

- 3) Transferring chemical parameters from Cu²⁺ to Zn²⁺ chemistry is very “bold”. The chemistry of the metal ions is quite different and I propose that the authors were just lucky to find in the diverse set of variables one hit. Thus capturing of chemical intuition by machine learning for a given chemical systems is really good, but the transferability is very limited.

Authors reply:

The reviewer confirms the point we aim to make in our manuscript and that the detailed synthesis conditions **cannot** be transferred to Cu²⁺ and Zn²⁺. But the point we also try to make is that from the failed experiments we obtained insights on the relative importance of the different parameters to impact the formation of crystals. With this information one can search significantly more efficient, but, indeed, one has to be lucky as well, we fully agree, there is no guarantee it will work for every case. In this way, our “intuition” is by no means different from the intuition developed by chemist in the lab; it is essential in many cases but there are some cases the chemistry can be surprisingly different.

To emphasize this point, we have added in the revised manuscript.

“The “intuition” we have quantified by machine learning is by no means different from the intuition developed by chemist in the lab; it is useful in many cases, but one always need to keep in mind that in some cases the chemistry can be surprisingly different.”

- 4) I fully agree with the authors that chemists need to report also the “failed” experiments, i.e. the ones that did not lead to the desired product. This gives valuable information for synthetic chemists trying to reproduce the results in another laboratory. But a very important problem, that is not covered in the manuscript, is the fact that the many more variables are influencing the properties of the product. Especially the size of the reactor employed but also things like purity of the starting materials or solvents is very important.

Authors reply:

We agree with the reviewer that many more synthesis variables play role. In current study we kept many variables fixed, e.g. size of reactor, purity of reactants, etc. Our methodology can, in principle, deal with these additional variables, but for each additional variable we need additional data. Our expectation is that if we involve different laboratories, we also get access to more data that will (partly) mitigate these concerns. In the revised manuscript we have added these examples of variability mentioned by the reviewer.

minor points

- 1) The reactions were not carried out in parallel (but serial), this is not possible with the microwave oven employed in the study.

Authors reply:

It was corrected in the revised version.

- 2) Fig. S6: please correct “Powder x-rad diffraction”

Authors reply:

It was corrected in the revised version.

- 3) “Machine learning: capturing chemical intuition” The first sentence is not true any more. Nowadays MOF scientist are more interested in other topics such as the properties, the up-scaling or the how the MOFs are formed. The search for “world record” BET is from yesterday. The expression “world record BET” does not make any sense (BET-value, BET theory, ...).

Authors reply:

We have modified the text, so it now reads “high experimentally measured BET value”. We partly agree with the reviewer. We agree that from a scientific point of view getting the highest BET is yesterday’s problem. Yet, we were, of course, happy to see that our systematic approach did give the highest value to date. Indeed, this example does illustrate an important underlying practical issue that finding the synthesis conditions to find the optimal performance can require a lot of experiments.

4) Figure 2: scale bars are missing/not readable.

Authors reply:

The figures adapted in the revised version accordingly.

REVIEWERS' COMMENTS:

Reviewer#1

I have the read the responses from the authors. I still have one objection (see below) but otherwise I am in agreement with all other points and think the revised manuscript has significantly improved.

My remaining objection: I still do not see the point of having tick marks on a plot where the axes are not labeled and where numbers are not shown. What are the point of the tick marks? If you agree that the tick marks serve no purpose, and that numbers are not needed for these figures, then the authors agree that these figures serve only to give a qualitative sense of the data. It is misleading, I think, to show qualitative data and make it appear as though it is quantitative. Alternatively, label and quantify the axes! Perhaps the authors disagree -- I will not force the issue.

Response to Reviewer #1

I have the read the responses from the authors. I still have one objection (see below) but otherwise I am in agreement with all other points and think the revised manuscript has significantly improved.

My remaining objection: I still do not see the point of having tick marks on a plot where the axes are not labeled and where numbers are not shown. What are the point of the tick marks? If you agree that the tick marks serve no purpose, and that numbers are not needed for these figures, then the authors agree that these figures serve only to give a qualitative sense of the data. It is misleading, I think, to show qualitative data and make it appear as though it is quantitative. Alternatively, label and quantify the axes! Perhaps the authors disagree -- I will not force the issue.

Authors reply:

We agree with the reviewer about the tick marks, and therefore, we removed the tick marks in the MDS plots in the revised version.